# Tuning Functional Behavior of Humic Acids through Interactions with Stöber Silica Nanoparticles

**DOI:** 10.3390/polym12040982

**Published:** 2020-04-23

**Authors:** Giulio Pota, Virginia Venezia, Giuseppe Vitiello, Paola Di Donato, Valentina Mollo, Aniello Costantini, Joshua Avossa, Assunta Nuzzo, Alessandro Piccolo, Brigida Silvestri, Giuseppina Luciani

**Affiliations:** 1Department of Chemical, Materials and Production Engineering, University of Naples “Federico II”, p.le V. Tecchio 80, 80125 Naples, Italy; giuliopota@gmail.com (G.P.); virginia.venezia@unina.it (V.V.); giuseppe.vitiello@unina.it (G.V.); anicosta@unina.it (A.C.); luciani@unina.it (G.L.); 2CSGI, Center for Colloids and Surface Science, via della Lastruccia 3, 50019 Florence, Italy; 3Department of Science and Technology, University of Naples “Parthenope”, Centro Direzionale Isola C4, 80143 Naples, Italy; pdidonato@uniparthenope.it; 4Center for Advanced Biomaterials for Health Care, Istituto Italiano di Tecnologia@CABHC, Largo Barsanti e Matteucci 53, 80125 Naples, Italy; Valentina.Mollo@iit.it; 5Laboratory for Biomimetic Membranes and Textiles, Empa, Swiss Federal Laboratories for Materials Science and Technology, Lerchenfeldstrasse 5, CH-9014 St. Gallen, Switzerland; joshua.avossa@unina.it; 6Centro Interdipartimentale di Ricerca sulla Risonanza Magnetica Nucleare per l’Ambiente, l’Agroalimentare ed i Nuovi Materiali (CERMANU), University of Naples “Federico II”, Via Università 100, 80055 Portici, Italy; assunta.nuzzo@unina.it (A.N.); alpiccol@unina.it (A.P.)

**Keywords:** hybrid nanoparticles, humic acid, silica nanoparticles, APTS, antioxidant properties

## Abstract

Humic acids (HA) exhibit fascinating multifunctional features, yet degradation phenomena as well as poor stability in aqueous environments strongly limit their use. Inorganic nanoparticles are emerging as a powerful interface for the development of robust HA bio-hybrid materials with enhanced chemical stability and tunable properties. Hybrid organic-inorganic SiO_2_/HA nanostructures were synthesized via an in-situ sol-gel route, exploiting both physical entrapment and chemical coupling. The latter was achieved through amide bond formation between carboxyl groups of HA and the amino group of 3-aminopropyltriethoxysilane (APTS), as confirmed by Fourier-Transform Infrared (FTIR) and Nuclear Magnetic Resonance (NMR) spectroscopy. Monodisperse hybrid nanoparticles about 90 nm in diameter were obtained in both cases, yet Electron Paramagnetic Resonance (EPR) spectroscopy highlighted the different supramolecular organization of HA. The altered HA conformation was reflected in different antioxidant properties of the conjugated nanoparticles that, however, resulted in being higher than for pure HA. Our findings proved the key role of both components in defining the morphology of the final system, as well as the efficacy of the ceramic component in templating the HA supramolecular organization and consequently tuning their functional features, thus defining a green strategy for bio-waste valorization.

## 1. Introduction

Bio-waste management and valorization is one of the most challenging issues for sustainable development, due to the enormous amount of bio-residues resulting from biogenic and non-biogenic transformations [1,2]. The great abundance and chemical richness of bio-waste make their valorization one of the most promising approach for moving toward a circular economy. In this field, nanotechnologies hold a key role in defining processes and innovative solutions to allow the conversion of residues into added-value nanostructured materials, with relevant properties for a huge range of applications, from medicine, to sensing, packaging, environmental protection and electronics [1,2]. Among bio-wastes, humic acids (HA) are the alkali-soluble fraction of natural organic matter, consisting in a multitude of heterogeneous organic molecules surviving the biological and chemical degradation of both vegetal and animal biomasses [3,4]. The associations of the amphiphilic HA components are stabilized by weak hydrophobic, hydrogen and metal-bridged electrostatic bonds in supramolecular architectures [5]. 

HA are considered as a promising and inexpensive source for high-value products and novel materials through the green chemistry approach [1,6]. The chemical heterogeneity and the metastable conformation of humic matter, as well as their different reactive functional groups, are responsible for a wide range of useful properties, including the adsorbing capacity towards metals and organic pollutants [7,8,9,10,11,12,13]. Moreover, they exhibit a strong attitude toward Reactive Oxygen Species (ROS) quenching, due to their intrinsic paramagnetic properties, resulting as effective antioxidant agents. The red-ox behavior of humic substances is strongly related to the phenol/quinone moieties in their complex supramolecular structures [14,15]. However, their rapid conformational dynamics and, hence, fast reactivity in water environments render humic substances of poor use for several applications in aqueous media [7,16].

A suitable solution may rely on the immobilization/encapsulation of HA onto an organic either inorganic support, providing for mechanical stability and steady reactivity. The embedding matrices act as a physical barrier to oxygen and other moieties, thus reducing HA oxidation processes and prolonging their shelf-life [17]. Inorganic nanoparticles are emerging as a powerful interface for the development of robust bio-hybrid materials with enhanced chemical stability and tunable properties [18]. In fact, the combination of HA with magnetic nanoparticles allows for their easy reuse and recycling and provides effective environmental solutions to heavy metals remediation from water [19]. 

Recent studies proved that the inorganic phase was able to tune bio-polymeric supramolecular structures, boosting their intrinsic properties and tuning their overall biological activity [20,21,22,23]. Silica has been identified as an ideal support for this method, due to its strong hydrophilicity, acknowledged biocompatibility, as well as tunable size, shape, porosity and surface chemistry [21,24]. The immobilization of HA onto silica appears as a valid approach to exploit the large reactive potential of these moieties for several technological solutions spanning separation chromatography [25], water remediation [9,26,27,28] and antioxidant products [14,15].

Two different strategies have been usually explored to combine HA with a silica support. The physical entrapment of humic molecules into silica matrices provides a simple but effective route to obtain HA-SiO_2_ composites with a high sorption capacity towards heavy metals cations [26] as well as organic hydrophobic contaminants [27]. Alternatively, chemical immobilization has been exploited to bind humic acids onto the silica surface [28,29,30,31]. 

Generally, the first step for the immobilization is the surface functionalization of the silica support with amino groups by silanization with 3-aminopropyltriethoxysilane. Binding can be carried out by electrostatic bonds (silylation) or by chemical coupling, thereby unavoidably involving the functional groups responsible for the HA multiple reactivity [31,32,33]. Therefore, both physical and chemical methods can influence the conformation of the HA supramolecular structure, thus defining the availability of reactive centers and ultimately determining the HA reactivity. 

In this scenario, this study aims at elucidating the influence of physical entrapment and chemical coupling strategies on the physical-chemical properties of HA-SiO_2_ nanomaterials.

To this purpose, two types of silica–HA hybrid nanoparticles were designed and prepared through an in-situ one-pot sol-gel route based on the modified Stöber method. Notably, the former (SiO_2_/HA_p) was obtained through hydrolysis and polycondensation of tetra propyl orthosilicate in the presence of HA in the reacting mixture. The latter (SiO_2_/HA_c) was prepared by exploiting 3-aminopropyltriethoxysilane as the coupling agent for HA, to allow chemical immobilization. 

Both synthesized samples were submitted to an in-depth physical–chemical characterization, integrating different techniques, such as Scanning Electron Microscopy (SEM) to assess the morphology and average size of the obtained nanoparticles, and BET porosimetry to evaluate the specific surface area and pore size distribution. Furthermore, Thermogravimetric Analysis (TGA) provided the amount of organic content in nanohybrids, while chemical immobilization was investigated through Fourier Transform Infrared Spectroscopy (FT-IR) and Nuclear Magnetic Resonance (NMR). 

Electron Paramagnetic Resonance (EPR) spectroscopy gave information about the presence of HA intrinsic radical species and their distribution within the nanostructures. The antioxidant potential of hybrid nanostructures was finally tested through the ferrous oxidation–xylenol orange FOX assay. Our results proved the key role of both components in defining the morphology of the final hybrid system, as well as the efficacy of the ceramic component in templating the supramolecular organization of HA, consequently driving functional properties and setting a green strategy for bio-waste valorization.

## 2. Experimental Section

### 2.1. Materials 

Tetrapropyl orthosilicate (TPOS, 99.999%), 3-aminopropyltriethoxysilane (APTS, 99.999%), 1-ethyl-3-(-3-dimethylaminopropyl) carbodiimide hydrochloride (EDC), 2-propanol (IPA), Ammonium hydroxide (NH_4_OH 33 %wt solution), and water from Sigma-Aldrich (Milan, Italy) were used as received. Humic acids were extracted from a mature compost made with manure (HA), as described elsewhere [34]. 

### 2.2. Synthesis of Humic Acid Functionalized Silica Nanoparticles

Humic Acid functionalized silica nanoparticles were prepared in situ following two different sol-gel routes. The former was based on the physical entrapment of HA superstructures during the synthesis of the silica phase, and the other one followed a chemical addition between the two phases [21]. 

In the first synthesis route, an aqueous solution of HA (2.7 mg/mL) was prepared. After the complete solubilization of HA, proper amounts of ammonia (120 μL) and 2-propanol (13 mL) were added, followed by the dropwise addition of TPOS (1.0 mL). The obtained system was kept 2 h at room temperature, and the HA-functionalized silica particles were centrifuged and washed three times with distilled water. These nanoparticles will be indicated in the following as SiO_2_/HA_p. Bare silica nanoparticles were produced following the same synthetic way, without the addition of HA, and they will be named as SiO_2__p.

In the second synthesis procedure, the APTS-HA hybrid precursor was first prepared following the EDC chemistry. EDC (6.0 mM) and APTS (12.0 mM) were added to the aqueous solution of HA. The reaction was allowed to proceed under stirring for 18 h. After this time, ammonia (120 μL), 2-propanol (13 mL) and TPOS (1.0 mL) were added to the solution, and the synthesis of hybrid nanoparticles was initiated. After 2 h of reaction, the nanoparticles were centrifuged and washed three times with distilled water; they will be noted as SiO_2_/HA_c. Bare silica nanoparticles were synthesized following the same procedure in the absence of HA, and this sample will be denoted as SiO_2__c.

### 2.3. Characterization Techniques

The morphological investigation of all nanoparticles was performed using both Scanning Electron Microscopy (SEM) and Transmission Electron Microscopy (TEM) using a Field Emission Ultrapluss ZEISS Microscope (ZEISS, Oberkochen, Germany) and FEI Tecnai G2 20 Microscope (FEI, Hillsboro, OR, USA). For the TEM investigation, 15 μL of Nanoparticles (NPs) suspension was spread on a copper grid (200 mesh with a carbon membrane). For the SEM investigation, a drop of NPs suspension was deposited onto the surface of an aluminum stub and then sputter-coated with a Pt/Pd layer (5 nm) in a Cressington sputter coater 208HR.

Zetasizer Nano Series (Malvern Instruments, Malvern, UK), using the laser dynamic scattering (*λ* = 632.8 nm) and the particle electrophoresis techniques, was used to carry out the size distribution and ζ-potential measurements. All the samples were diluted in distilled water, and five runs (lasting 100 s) were used with a detecting angle of 173° in the calculations of the particle diameter distribution. The ζ-potential measurements were carried out by setting 50 runs for each measurement.

A Nicolet Instrument Nexus model (Thermo Scientific, Waltham, MA, USA)equipped with a DTGS KBr (deuterated triglycinesulfate with potassium bromide windows) detector was used to perform the FT-IR investigation. IR absorption spectra of all samples were recorded in the 4000–400 cm^−1^ range at a 2 cm^−1^ resolution on pressed disks of the powders previously diluted in KBr (1 wt %). The IR spectrum of each sample was corrected for the spectrum of blank KBr.

The thermal behavior of samples was evaluated by a TA Instrument simultaneous thermoanalyser SDT Q600 (TA Instrument, New Castle, DE, USA) A Thermogravimetric Analysis (TG) were performed on 10 mg dried samples in N_2_, from 25 to 900 °C, with a heating rate of 10 °C/min. 

The solid-state ^13^C-CPMAS-NMR spectrum of pure HA was acquired with a 300 MHz Bruker Avance wide-bore magnet (Bruker Bio Spin GmbH, Rheinstetten, Germany), equipped with a CPMAS probe, while the spectra of the HA-APTS hybrid precursor were obtained with a 400 MHz Bruker Avance magnet (Bruker Bio Spin GmbH, Rheinstetten, Germany), equipped with a ^1^H-^13^C HRMAS probe. For the CPMAS spectroscopy, the sample was loaded into 4-mm zirconia rotors, closed with Kel-F caps and spun at a rate of 13,000 ± 1 Hz. The spectrum was acquired by applying a cross-polarization technique and consisted of 1814 time-domain points, a spectral width of 300 ppm (22,727.3 Hz), a recycle delay of 2 s, 5000 scans and 1 ms of contact time. The ^13^C-CPMAS pulse sequence was conducted by using a ^1^H RAMP pulse to account for the *non*-homogeneity of the Hartmann–Hahn condition. A TPPM15 scheme was applied to perform the ^13^C-^1^H decoupling. The Free Induction Decay (FID) was transformed by applying a 4k zero filling and an exponential filter function with a line broadening of 100 Hz. The NMR spectrum was acquired at a temperature of 298 ± 1 K and processed by using MestReC NMR Processing Software (v.4.8.6.0, Cambridgesoft, Cambridge, MA, USA). Zero filling was applied during the Fourier transform of free induction decays (FIDs). For the HRMAS probe, about 15 ± 2 mg of the HA-APTS sample was packed into a HRMAS-NMR 4 mm zirconia rotor, fitted with a perforated Teflon insert, soaked with approximately 15 μL of D2O solution (99.8% D_2_O/H_2_O, Armar Chemicals) and sealed with a Kel-F cap (Rototech-Spintech GmbH, Bad Wildbad, Germany). The 2D hetero-nuclear experiment, HSQC (Heteronuclear Single Quantum Coherence) was performed to identify the ^1^H-^13^C correlations and assign the most intense NMR signals detected in the samples. The HSQC experiment was acquired with a spectral width of 16 (6410.3 Hz) and 300 (30,186.8 Hz) ppm for 1H and 13C nuclei, respectively, a time domain of 2048 points (F2) and 256 experiments (F1), 16 dummy scans, 80 total transients and 0.5 μs of trim pulse length. The experiment was optimized by considering 145 and 6.5 Hz as the optimal ^1^H-^13^C short- and long-range J-couplings, respectively. Spectra were processed by using both Bruker Topspin Software (v 2.1, BrukerBiospin, Rheinstetten, Germany) and MNOVA Software (v.9.0, Mestrelab Research, Santiago de Compostela, Spain). A two-fold zero-filling was applied during the Fourier transformation of free induction decays, whereas the apodization function was not necessary. Phase and baseline corrections were conducted on bi-dimensional spectra. ^13^C and ^1^H NMR frequency axes were calibrated by associating both carbon and proton peaks of the α-Glucose anomeric group to 94.73 and 5.22 ppm, respectively [35].

The Electron Paramagnetic Resonance (EPR) analysis was carried out by using an X-band (9 GHz) Bruker Elexys E-500 spectrometer (Bruker, Rheinstetten, Germany), equipped with a super-high sensitivity probe head. Samples were transferred to flame-sealed glass capillaries which, in turn, were coaxially inserted in a standard 4 mm quartz sample tube. Measurements were performed at 25 °C. The instrumental settings were: sweep width, 100 G; resolution, 1024 points; modulation frequency, 100 kHz; and modulation amplitude, 1.0 G. The amplitude of the field modulation was preventively checked to be low enough to avoid a detectable signal overmodulation. EPR spectra of all analyzed samples were recorded with a microwave power of 0.06 mW to avoid the microwave saturation of the resonance absorption curve and by accumulating 32 scans to improve the signal-to-noise ratio. Power saturation curves were also recorded by varying the microwave power from 0.004 mW to 128 mW. Finally, the quantitative analysis of EPR spectra, in terms of g factor and spin-density, was realized by inserting an internal standard, composed of Mn^2+^/MgO powder, in the quartz tube co-axially with each sample.

### 2.4. Ferrous Oxidation Xylenol Orange (FOX) Assay

SiO_2_, SiO_2_-humic acids nanoparticles and free HA suspensions in distilled water (0.04 mg/mL) were incubated in the presence of H_2_O_2_ 25.0 μM, 12.5 μM or 10.0 μM at room temperature, under stirring. Aliquots (100 μL) were withdrawn at times 0′, 30′, 1 h and 4 h. Each aliquot was then mixed with 900 μL of reagent mixture containing 0.10 mM xylenol orange, 0.25 mM ammonium iron(II) sulfate hexahydrate in 250 mM H_2_SO_4_ (10 mL) and 3.88 mM 1,1,3,3-tetramethoxypropane, butylated hydroxyanisole (BHA) in methanol (90 mL) [36]. After 30 min under stirring at room temperature, the absorbance at 593 nm was measured. Control experiments were performed in the absence of nanoparticles and by using non-bound HA in the same concentration. The antioxidant activity of all the samples was expressed as the residual percentage of H_2_O_2_ measured in the same conditions in the absence of nanoparticles.

## 3. Results and Discussion

Hybrid HA/Silica nanoparticles were obtained by a sol-gel methodology which relies on nanoparticles’ formation in the presence of HA (in-situ route). Two synthetic approaches were followed: the former based on a physical entrapment of HA superstructures and the latter based on a chemical coupling between the organic and inorganic phases [37,38].

The morphological investigation of all synthesized nanoparticles was performed through Scanning Electron Microscopy (SEM), as reported in Figure 1. 

The SEM images of the bare SiO_2__p sample (Figure 1A) revealed nanoparticles with a spherical shape and a narrow size distribution of about 400 nm in diameter. A bimodal size distribution was clearly observed in the bare SiO_2__c sample (Figure 1B) made of two nanoparticle populations: the former composed by few large nanoparticles (400 nm in diameter) and the latter made up of a significant number of small nanoparticles (100 nm in diameter). Actually, the SiO_2__p sample was prepared following a modified Stöber method [39]. As expected, the use of only one alkoxide combined with appropriately chosen amounts of reagents allows for deep control over the nucleation and growth processes, leading to spherical and highly monodisperse nanoparticles. The widely recognized mechanism for Stöber silica NPs formation is reported in Scheme 1 (route A) [39,40,41,42]. 

First, hydrated monomers obtained from the hydrolysis of precursors undergo condensation reactions, forming silicate polymer chains and then small nuclei. These grow through silica monomer- and polymer bonding, forming primary particles, usually 5–7 nm in mean diameter [41,43]. Then, primary particles aggregate, producing larger SiO_2_ particles that grow up to a stationary critical size, able to generate a double electrical layer that hinders the further aggregation of primary particles onto their surface. Starting from one monomeric precursor, such as TPOS, brings a uniform size distribution of nuclei and subsequently of primary particles, finally providing a narrow size distribution of nanoparticles. This mechanism leads to monodisperse particles [40,41,44,45].

Two different pathways are expected to coincide for the formation of SiO_2__c nanoparticles. On one hand, the pre-hydrolysis of APTS is supposed to form a certain number on nuclei, which should act as binding sites for hydrolyzed TPOS moieties and lead to the fast formation of big nanoparticles. On the other hand, residual hydrolyzed TPOS, not involved in the previous process, must give rise to a different population of nuclei and subsequently of primary particles, which are thermodynamically more likely to aggregate into new secondary particles rather than join the surface of big nanoparticles. This process should give rise to a second wide population of smaller nanoparticles than SiO_2__p samples, as expected for a lower concentration of TPOS. The overall process results in a bimodal particle size distribution, as revealed from SEM micrographs (Figure 1B) [40].

SEM micrographs of both SiO_2_/HA hybrid nanoparticles (Figure 1C,D) revealed a narrow size distribution of pseudospherical particles of about 90 nm in mean diameter, lower than bare SiO_2_ (SiO_2__p, SiO_2__c). These results were confirmed by TEM micrographs reported in Figure 2, also revealing an intimate mixing of both components, as no differences in contrast were observed. Furthermore, at a high magnification the hybrid nanoparticles clearly showed a cluster architecture consisting of smaller primary particles, in accordance with the proposed formation mechanism of particle growth through aggregation [41].

These clusters could also be formed through the previously reported intermolecular interactions among HA molecules [46].

The nanoparticle size of both hybrid samples was also confirmed by DLS investigations. Figure 3 reports the size distribution of both SiO_2_/HA_p and SiO_2_/HA_c nanoparticles.

The hydrodynamic diameters of SiO_2_/HA_p and SiO_2_/HA_c NPs were about 97 nm (PDI = 0.155) and 105 nm (PDI = 0.183) respectively, and appeared slightly larger than the dry particle diameters observed through TEM analysis. Furthermore, the ζ-potential values were −36 ± 8 mV and −40 ± 7 mV respectively, showing no significant surface charge differences between the two systems. Thus, the electrostatic repulsion kept both HA functionalized nanoparticles stably dispersed in water. 

Actually, both synthetic routes were based on the formation of silica particles in an HA-containing solution. Both morphological and DLS investigations revealed that the HA phase actively participated in the nanoparticles’ formation, controlling their growth. Notably, HA superstructures limited the diffusion of both silica oligomers and nanoparticles in solution. Furthermore, the adsorption of HA moieties onto nanoparticles results in a highly negatively charged surface which can hamper their aggregation due to electrostatic repulsions [47]. Overall, these phenomena resulted in a smaller size of hybrid nanoparticles than for the bare SiO_2__p and SiO_2__c samples. 

The FT-IR investigation of both SiO_2_/HA_p and SiO_2_/HA_c hybrid NPs, reported in Figure 4, revealed the main characteristic bands of the silica phases. The band at ~3500 cm^−1^ was assigned to the OH stretching vibration of surface silanol groups and absorbed water. The bands at 1100 cm^−1^ were attributed to the Si–O–Si stretching vibration modes in SiO_4_ units, while the band at 950 cm^-1^ was related to the non-bridging Si–O stretching vibration. The bands at 800 and 470 cm^−1^ were attributed to the Si–O–Si stretching vibration between two adjacent tetrahedra and to the Si–O–Si bending, respectively. The assignments were reported in Table 1 [48]. 

In order to confirm the presence of the HA component in the hybrid system, Thermogravimetric Analysis (TGA) measurements were carried out, and the thermal behavior of the SiO_2_/HA samples was compared with those of pure HA and bare SiO_2_ nanoparticles. The resulting TGA curves are reported in Figure 5. The TG curve of pure HA showed a first weight loss below 150 °C (10 wt %) related to the release of physically adsorbed water, and that above 250 °C was attributed to the oxidative decomposition of both labile and recalcitrant components [31].

Furthermore, a total mass loss of approximately 80 wt % was evaluated. 

The TG curve of both SiO_2__c and SiO_2__p samples showed a low temperature weight loss (about 7%) in the range of 25–120 °C, related to the removal of physically adsorbed water. The second weight loss, in the range 250–600 °C, was attributed to the degradation of alkyl chains, more pronounced in the SiO_2__c sample with the non-hydrolyzable aminopropyl group, and the dehydroxylation of Si–OH residual groups, forming Si–O–Si groups.

The TG curves of both SiO_2_/HA_p an SiO_2_/HA_c showed an additional visible weight loss in the range of 200–580 °C, related to the decomposition of HA. A different amount of HA was estimated in the hybrid NPs: 15 wt % in the case of SiO_2_/HA_p NPs and 30 wt % of HA for SiO_2_/HA_c NPs. This result suggests that a chemical coupling between the two phases resulted in a greater amount of HA in the final system. 

Furthermore, no differences in the thermal degradation behavior of HA were perceived for both hybrid samples, since no changes were observed in either the starting degradation temperature or in the inflection point. 

The porosity of the hybrid nanoparticles was assessed by means of N_2_ adsorption/desorption measurements (data not shown). The BET analysis revealed a very similar surface area for both bare SiO_2_ NPs of about 20.10 m^2^/g. A slight increase of the surface area was observed in both SiO_2_/HA_p and SiO_2_/HA_c NPs: 33.54 and 50.48 m^2^/g, respectively. This was attributed to the presence of humic components, which usually show a highly narrow microporosity [49,50].

In order to investigate the chemical coupling between the two components, the HA_APTS hybrid precursor, produced during the first step of the synthesis of the SiO_2_/HA_c sample, was analyzed. The FT-IR spectra of APTS, bare HA and the HA_APTS hybrid monomer were carried out and are reported in Figure 6. The typical HA FT-IR spectrum (black curve) revealed a broad adsorption band in the region of 3700–2500 cm^−1^ attributed to the –OH stretching vibration, and the bands in the range from 2800 to 3200 cm^−1^ were attributed to the C–H symmetric and asymmetric stretching vibrations of alkyl structures. Additional bands are also visible in the region of 1580–1660 cm^−1^, related to the C=C bond in aromatics and olefins as well as to the carboxyl C=O bond, ketone and quinone groups, whereas the band at 1514 cm^−1^ was attributed to the ring vibrating modes of ortho-substituted aromatic compounds. The bands around 1280–1070 cm^−1^ corresponded to the stretching vibration of phenolic C–O and aliphatic OH. Finally, the bands in the region of 1460-1390 cm^-1^ corresponded to the OH of the phenols, COO– and –CH_3_ bending vibration mode [44,51,52]. 

The FT-IR spectrum of the HA-APTS hybrid precursor showed a marked decrease of the bands in the region between 1400 and 1650 cm^−1^, including the COO^-^ asymmetric stretching vibration modes [53,54,55]. Furthermore, the spectrum showed a shoulder at around 1560 cm^−1^, which can be assigned to the out-of-phase combination mode of the N–H in the plane bend and the C–N stretching vibration of the Amide II band [56,57]. Finally, by comparing the FTIR spectra of the bare HA and HA-APTS precursor, changes in the adsorption band of the OH vibrations in the range of 3500 cm^−1^ were clearly visible. The differences in this region were attributed to the N–H stretching in the amide groups [55,58]. Furthermore, the changes in the range of 3500 cm^−1^ and weak bands in the 1400–1600 cm^−1^ region were also visible in the FTIR spectrum of the SiO_2_/HA_c sample (Figure 4), confirming the presence of HA in hybrid nanoparticles [59]. 

In order to further support the FT_IR results, an NMR investigation was carried out. Figure 7 reports the ^13^C CPMAS NMR spectra of pure HA (Figure 7A) and the HA-APTS hybrid precursor (Figure 7B).

The total area of the bare HA spectrum was considered as 100% and divided into six regions (i.e., 190–160, 160–145, 145–110, 110–60, 60–45 and 45–0 ppm). The percentage of each resonant carbon nucleus was reported in Table 2. 

In Figure 7B, a reduction of the peak at 173 ppm, related to the carboxyl groups region, was clearly visible, suggesting an involvement of these groups in the reaction with APTS. 

A comparison of the results obtained by ^13^C CP/MAS spectrum and two-dimensional ^1^H-^13^C HSQC, reported in Figure 8, revealed the presence of a signal at 170/9 ppm, supporting the formation of an amide bond between the carboxyl groups of HA and amine groups of APTS, in accordance with the FTIR results (Figure 6).

The EPR analysis of hybrid SiO_2_/HA nanoparticles was performed according to the procedure recently used in the characterization of melanin-like nanomaterials [20,21,22], to provide important information about the nature of radical species as well as the supramolecular properties of the humic acid moiety forming the hybrid SiO_2_-based material. The EPR spectra of the hybrid samples are reported in Figure 9, whereas the corresponding spectral parameters are summarized in Table 3. The pure HA was also analyzed as a reference.

The EPR spectra shown in Figure 9 showed a similar line shape, formed by a single and roughly symmetric peak at a g-factor of ~2.0035, typical of carbon-centered radicals of polyaromatic molecules, as reported in the literature [20,60]. By comparing the EPR spectra, no significant differences were observed in the line shape of the SiO_2_/HA_p sample with respect to that of the pure HA. Indeed, this evidence was validated by the quantitative determination of the signal amplitude, ΔB, which is related to the line width of the EPR peak, directly measurable by the experimental spectra (see Figure 9A). 

The line width of the EPR signals reflects the relaxation time of spinning electrons and is primarily affected by the unresolved hyperfine interaction between unpaired electrons with the neighboring atoms [60,61]. Consequently, it can furnish information about the presence of different carbon-based radical species, as well as about the supramolecular organization of polyaromatic structures hosting unpaired electrons. For this reason, the ΔB parameter is usually considered indicative of the mean distance between the radical centers. As reported in Table 3, the determined ΔB value of the SiO_2_/HA_p sample was quite similar to that of pure HA, suggesting that the chemical and structural organization of the organic moiety was not particularly perturbed after combination with the inorganic matrix. 

On the other hand, a great ΔB increase (~1.2 G) was observed for the SiO_2_/HA_c sample, indicating that the chemical coupling, due to the APTS use during the synthesis, strongly influenced the local organization of the humic macromolecules, in which the radical centers appeared closer within the hybrid nanoparticles. This behavior was also confirmed by the power saturation curves, reported in Figure 9B and obtained by plotting the normalized peak amplitude (A/A0) of spectra recorded while increasing the incident microwave power (P) as a function of the square root of P itself. In all cases, a monotonic trend was observed, indicating that the free radical spins did not exhibit the same relaxation times and saturated independently.

This behavior is a mark of a chemical inhomogeneous distribution of radical centers characterizing the HA component, and probably associable with the presence of different aromatic moieties on which the unpaired electrons were stabilized. More specifically, no changes were observed in the case of the PS curve of SiO_2_/HA_p with respect to pure HA.

On the other hand, a decrease in the inhomogeneity character was detected in the PS curve of the SiO_2_/HA_c sample, supporting the idea of changes in the HA supramolecular organization, in which the organic component tended to be more confined within the hybrid nanoparticles, as a consequence of a different growth mechanism induced by the chemical conjugation of HA macromolecules with APTES and its subsequent combination with the SiO_2_ matrix. Actually, during silica formation on APTS nuclei, HA hydrophobic chains tend to interact and segregate in the interior of the born particles [31].

Finally, the spin-density values demonstrated a higher concentration of radical species in the SiO_2_/HA_c sample with respect to the SiO_2_/HA_p one, confirming the presence of a higher organic content, in agreement with the TG analysis.

To assess the effect of immobilization onto the silica phase, as well as of different combination strategies on the functional behavior of HA, the antioxidant potential of the nanoparticles was evaluated by means of the FOX assay, i.e., the ferrous ions oxidation in the presence of the ferric ions indicator xylenol orange. The FOX assay is currently used to detect the levels of hydroxyperoxides in different kinds of matrixes. The assay allows one to measure the ability of a compound to act as an antioxidant by causing a decrease in the levels of peroxides like H_2_O_2_.

The antioxidant properties of both hybrid systems (SiO_2_/HA_p and SiO_2_/HA_c NPs) were evaluated in the presence of H_2_O_2_ and comparing them with those of the free acid (HA) and bare SiO_2_ (the reported results were representative of both SiO_2__c and SiO_2__p). The antioxidant potential was measured by adding hydrogen peroxide at three concentrations of, i.e., 25, 12.5 and 10 μM: the results obtained are shown in Figure 10A–C, respectively. The standard calibration curve of the absorbance plotted versus the H_2_O_2_ respective concentrations is also reported in Figure 10D.

For all the H_2_O_2_ concentrations used and for all the time intervals, the highest antioxidant potential was shown by the SiO_2_/HA_p sample; moreover, both the NP samples SiO_2_/HA_p and SiO_2_/HA_c were shown to be more active than the free HA tested with the same weight amount of nanoparticles. Notably, the amount of HA in the hybrid systems was lower than for the tests that employed it as a free acid, thus confirming that the inclusion of the HA in the NP systems enhanced its antioxidant properties, thus requiring lower amounts of material to exploit its action.

Furthermore, physical entrapment (SiO_2_/HA_p sample) ensured a higher radical scavenging activity than chemical coupling. As widely assessed, the red-ox behavior of these mixtures was strongly related to the phenol/quinone moieties of their complex molecular structures. Indeed, these functional groups can act as electron donors: after losing one electron per stage, they span from a hydroquinone to a quinone state. In SiO_2_/HA_c samples, these species are confined to the interior of silica due to chemical coupling with APTS. These features probably decreased the availability of these reactive species, ultimately reducing the antioxidant activity.

## 4. Conclusions

This study reports the synthesis of hybrid SiO_2_/HA nanostructures via an in-situ sol-gel route, following two combination approaches between organic and inorganic components. Physical entrapment was achieved through the in-situ formation of silica nanoparticles in an HA-containing solution, whereas 3-aminopropyltriethoxysilane was exploited as chemical coupling agent between HA and the inorganic matrix through amide bond formation.

Our results prove that the combination with an inorganic component at the molecular scale is successful in addressing poor stability in aqueous solutions due to aggregation and degradation phenomena, while concurrently boosting the intrinsic antioxidant properties of HA. In fact, the HA components actively participated in nanoparticles’ formation, thus controlling their growth and leading to a smaller size than for bare SiO_2_ in the same synthesis conditions.

Furthermore, as shown by the FT-IR, NMR and EPR results, the combination strategy of humic biomaterials with an inorganic component has a great influence on the HA supramolecular organization, radical properties, as well as on the nature and availability of functional groups. These features ultimately impact on the antioxidant activity of final hybrid systems.

Notably, chemical coupling involves the reactivity of HA moieties and strongly influences the local organization of humic superstructures. This decreased availability of reactive centers towards free radical species resulted in a lower antioxidant activity than for hybrid systems obtained by physical entrapment of HA molecules.

These findings disclose the great potential of an inorganic templated approach as an effective strategy to turn bio-wastes into functional valuable materials.

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
