# Peer review of "Tuning Functional Behavior of Humic Acids through Interactions with Stöber Silica Nanoparticles"

_polymers, 2020, doi:10.3390/polym12040982_

Round 1
Reviewer 1 Report
I agree with the changes made by Giulio Pota, et al., of the manuscript “Tuning functional behaviour of Humic Acids through interactions with Stöber silica nanoparticles”, I think it is enough for publication in the journal "Polymers"
Author Response
The authors thank the reviewer for his positive comment.
Reviewer 2 Report
In this manuscript, both physical entrapment and chemical coupling are exploited to combine HA with inorganic silica nanostructure. The inorganic matrix is proved to influence the HA supramolecular structure as well as their reactivity, ultimately determining the activity of conjugated species. All my comments previously suggested at the first time this manuscript was submitted (polymers-724600), are successfully responded and resolved. This manuscript is now in good organization and is suggested to be accepted after minor revision.
- Corresponding reference should be cited to enrich the research background of interactions between molecules and inorganic nanomaterials. For example, Materials Today Communications 2020, 23, 101086; International Journal of Molecular Science 2019, 20(3), 741, et al
- Scheme 2 describes the normally accepted and widely known carboxyamine bond interactions induced by EDC. Based on this, there is no need to draw such a schematic diagram.
Author Response
- Corresponding reference should be cited to enrich the research background of interactions between molecules and inorganic nanomaterials. For example, Materials Today Communications 2020, 23, 101086; International Journal of Molecular Science 2019, 20(3), 741, et al.
The suggested references were added in the text as new Ref. 37 and 38 (Pag.5, line 213) and in the Reference list (Pag.17, line 585-590).
- Scheme 2 describes the normally accepted and widely known carboxyamine bond interactions induced by EDC. Based on this, there is no need to draw such a schematic diagram.
As suggested, the Scheme 2 has been deleted by the main text (Pag.10, line 345-350).
Reviewer 3 Report
All comments of this referee were addressed satisfactorily
Author Response
The authors thank the Reviewer for his positive comment.
This manuscript is a resubmission of an earlier submission. The following is a list of the peer review reports and author responses from that submission.
Round 1
Reviewer 1 Report
The information goes in the file

Reviewer 2 Report
In this manuscript, both physical entrapment and chemical coupling are exploited to combine HA with inorganic silica nanostructure. The inorganic matrix is proved to influence the HA supramolecular structure as well as their reactivity, ultimately determining the activity of conjugated species. This manuscript is in good organization and is suggested to be accepted after minor revision. The details are as below:
1. 13C-Cross Polarization Magic Angle Spinning NMR spectrum of HA in Table 1 and the corresponding relative C distribution in chemical shift regions (ppm) in 13C-CPMAS-NMR spectrum of MA in Table 1, is not the result closely related to the content discussed in the manuscript. These results are suggested to be deleted in the revised manuscript. Unless the authors add relevant discussion in the manuscript, for example, the comparison of the changes of the NMR results of HA before and after modification onto the surface of silica nanoparticles.
2. The authors state that, “these grow through silica monomer and polymer bonding forming primary particles, usually 5-7 nm in mean diameter.” The authors should supply the corresponding evidence for this statement. For example, the authors should give TEM images of the polymer bonding forming primary particles or cite corresponding references to support their statements.
3. According to Figure 4, should the clusters be formed through the intermolecular interactions between HA molecules? The authors should clarify this.
4. The content of Table 2 should be discussed more detailed according to the FTIR results in Figure 6, not just cite a reference. In addition, it is better for the authors to put Figure 5 and Figure 6 together as sub-figures, since these FTIR results represent the different stages of reaction.
5. The authors state that, “The line-width of EPR signals reflects the relaxation time of spinning electrons and is … of polyaromatic structures hosting unpaired electrons.” Corresponding references should be cited to support their statements.
6. Error bar should be given to the data points in Figure 8.
Reviewer 3 Report
The authors investigate humic acid incorporated silica nanoparticles for their particle size distribution, spectroscopic characterization and antioxidant effects as a green strategy for bio-waste valorization.
Comments:
- In the Introduction, the authors must discuss briefly bio-waste valorization and the role of nanoparticles in this process.
- In the modified Stober’s synthesis of silica nanoparticles, the use of cationic surfactants is avoided. How does anionic humic acid stabilize the silica nanoparticles from aggregation during synthesis?
- The zeta potential and particle size distribution should be characterized by dynamic light scattering (DLS).
- For the Ferrous oxidation xylenol orange (FOX) assay, the authors must present the data pertaining to the standard plot of absorbance versus μmole equivalents of hydrogen peroxide.
- In Figure 8, the antioxidant properties data shown in the line graphs must be shown with error bars.
- In the context of bio-waste valorization, the design of reusable, recyclable and recoverable nanoparticles such as magnetic nanoparticles are more environmentally better solutions. Also, humic acid has been previously demonstrated in heavy metals removal from water. Refer - Sci. Technol. 2008, 42, 18, 6949-6954.
- Statistical analysis must be performed for all the data shown in the line graphs. Please indicate statistical significance at p < 0.05.